# Parents' views on accepting, declining, and expanding newborn bloodspot screening

**Sylvia M. van der Pal**[1☯]*, **Sophie Wins**[1☯], **Jasmijn E. Klapwijk**[2], **Tessa van Dijk**[2], **Adriana Kater-Kuipers**[2], **Catharina P. B. van der Ploeg**[1], **Suze M. P. J. Jans**[1], **Stephan Kemp**[3], **Rendelien K. Verschoof-Puite**[4], **Lion J. M. van den Bosch**[5], **Lidewij Henneman**[2]

**1** Child Health, TNO, Leiden, The Netherlands, **2** Department of Human Genetics and Amsterdam Reproduction & Development Research Institute, Amsterdam UMC, Vrije Universiteit Amsterdam, Amsterdam, The Netherlands, **3** Department of Clinical Chemistry, Laboratory Genetic Metabolic Diseases, Amsterdam Neuroscience, Amsterdam Gastroenterology Endocrinology Metabolism, Amsterdam UMC, University of Amsterdam, Amsterdam, The Netherlands, **4** Department of Vaccine Supply and Prevention Programmes, National Institute for Public Health and the Environment (RIVM), Bilthoven, The Netherlands, **5** Centre for Population Screening, National Institute for Public Health and the Environment (RIVM), Bilthoven, The Netherlands

☯ These authors contributed equally to this work.
* sylvia.vanderpal@tno.nl

**Data Availability Statement:** The data underlying the results are available from the secretary of department of Child Health, TNO (contact via

## Abstract

### Introduction

The goal of newborn bloodspot screening (NBS) is the early detection of treatable disorders in newborns to offer early intervention. Worldwide, the number of conditions screened for is expanding, which might affect public acceptance. In the Netherlands, participation is high (>99%), but little is known about how parents perceive NBS. This study assessed parents' views on accepting, declining and expanding NBS.

### Methods

A total of 804 of 6051 (13%) invited parents who participated in NBS in the Netherlands during the last two weeks of December 2019, and 48 of 1162 (4%) invited parents who declined participation in NBS in 2019 and 2020, completed a questionnaire.

### Results

The most important reason for parents to participate in NBS was to prevent health complaints, whereas the most important reason to decline NBS was parents' viewpoint on life and the belief that the heel prick would be painful for the child. Compared to NBS participants, respondents who declined NBS were more actively religious, considered alternative medicine or lifestyle more important, were less inclined to vaccinate their child for infectious diseases, and reported more doubt about NBS participation (all differences p < .001). Informed choice was lower among respondents who declined NBS (44%) compared to participants in NBS (83%, p < .001), mostly due to insufficient knowledge. Of the NBS participants, 95% were positive about NBS expansion. Most NBS participants agreed to include

childhealthsecretary@tno.nl) for researchers who meet the criteria for access to confidential data.

**Funding:** This questionnaire study received funding from the Netherlands Organisation for Health Research and Development (ZonMw grant no. 543002006). The funders had no role in study design, data collection and analysis, decision to publish, or preparation of the manuscript. The authors received no other specific funding for this work.

**Competing interests:** The authors have declared that no competing interests exist.

conditions that could unintentionally reveal a diagnosis in the mother instead of the child (86%) or a condition that may not cause symptoms until later in the child's life (84%).

## Conclusion

Most participants made an informed decision to participate in NBS and are positive about screening for more conditions. Insights into parents' views on (non-)participation and expansion of NBS can help to ensure that NBS suits the population needs while safeguarding ethical principles for screening.

## Introduction

Newborn bloodspot screening (NBS) is considered one of the most successful public health programs, aiming at early detection of treatable disorders in newborns to offer early intervention [1]. Worldwide, due to technological developments, improved clinical understanding, and the availability of new treatments, the number of disorders eligible for NBS is continuously increasing [2, 3].

Studies have shown that the public is generally positive about NBS and its expansion [4–7]. In the Netherlands NBS participation is not mandatory and parents are asked if they agree to NBS [8]. NBS is nevertheless well accepted. In 2020, 99.4% of parents of newborns participated in newborn screening which, in that year, included 24 disorders [9]. Few studies have addressed parental reasons to decline NBS [10, 11]. More insight into the reasons for non-participation can lead to valuable knowledge that can be used, for example, to improve information for parents. It is important that parents make an informed decision to participate and maintaining a high acceptance and participation level is needed to achieve maximal health benefit.

Although expanded NBS can improve early diagnosis and treatment of more disorders to prevent irreversible health damage [12], it also raises questions, and is expected to increase the number of false positives and uncertain outcomes which may have far-reaching implications [13]. In the Netherlands, ethical issues challenge implementation of the addition of new disorders to the Dutch program as advised by the Dutch Health Council [14]. For example, the screening for organic cation/carnitine transporter 2 (OCTN2) deficiency in newborns may detect carnitine deficiency in asymptomatic mothers, which is beyond the goal of NBS and may have a medical and/or psychological impact on these women [15]. Furthermore, in 2015 the Dutch Health Council recommended screening only boys for X-linked adrenoleukodystrophy (ALD) [16]. With timely detection, ALD is treatable in boys, whereas symptoms in female carriers usually only develop later in life and are untreatable [17]. Studies show that most parents want to know whether their child has a certain condition, regardless of the availability of effective treatment [7, 18, 19]. However, it is unclear how screening under certain conditions, for example screening only boys, is perceived by parents.

Since societal acceptance of NBS is an important criterion for success, this study aimed to assess parents' reasons for, and views on participating or not participating in Dutch NBS, whether they make an informed decision, and their views on expansion of NBS in the Netherlands.

## Materials and methods

### Design and ethics statement

A cross-sectional survey study was performed using questionnaires among participants and non-participants in the Dutch NBS program. The Medical Ethical Committee of the VU University Medical Center Amsterdam approved the study protocol (no. 2019–509). Before parents completed the (online) questionnaire, they were asked to tick a box (written consent) to confirm that they had read the information letter and that they gave their informed consent to participate in the study.

### Setting

In the Netherlands, NBS is organized and coordinated by the Dutch National Institute for Public Health and the Environment (RIVM). Standardized information on NBS, including a leaflet, is provided during pregnancy by an obstetric healthcare provider, and issued before the screening is done, a few days after birth, at home by a youth health care worker, maternity nurse or midwife. When the baby is admitted to the hospital during the first week after birth, the blood spot is collected by a hospital health care worker. If the NBS result is good parents will get a letter from RIVM within five weeks, as from March 1st, 2020 (before that time, no result was given). If an abnormality is detected, the general practitioner will inform parents about the results.

### Study population

During the last two weeks of December 2019, all parents whose child had been offered NBS five weeks earlier (n = 6,051), received a letter from the RIVM inviting them to participate in a survey study. The invitation could only be sent by the RIVM due to of privacy reasons, and therefore no reminders could be sent. A period of five weeks was chosen because almost all parents will know their child's test result by then. In addition, all parents who declined participation in NBS in the whole of 2019 (n = 515) received an invitation to complete the same online questionnaire at the beginning of 2020, and all parents who declined participation in NBS in 2020 received an invitation at the beginning of 2021 (n = 647). When parents indicated that they did not participate in NBS, questions that were not relevant were automatically skipped. The questionnaire was available in Dutch and English and could be filled in online or could be sent to them on paper by post upon request. As the number of non-participants in 2019 who filled out the questionnaire was very low, all NBS non-participants in 2020 were sent a paper questionnaire, in addition to the online link, to lower the threshold for completing the questionnaire.

### Questionnaire and procedure

The questionnaire was developed based on interviews with stakeholders [14], a previous questionnaire study about Dutch parents' attitudes towards NBS [20, 21], and literature [19, 22]. The questionnaire was pre-tested among five parents. Readability of the questionnaire was evaluated by Pharos, the Dutch Centre of Expertise on Health Disparities. The questionnaire assessed reasons for participating and not participating, informed decision to participate or not, views on current NBS, views on expanding NBS, and background characteristics. NBS non-participants were not asked about their views on expanding NBS and views on current NBS to keep the questionnaire as short as possible and encourage completion. A copy of the entire questionnaire can be found in S1 Appendix. The different themes in the questionnaire are described below.

**Reasons to participate or not in NBS.** Respondents were first asked if they participated in NBS. Respondents were then asked to rate the importance of eight reasons to participate or not in NBS on a 5-point scale ranging from not at all important (1) to very important (5), and an open question was added to check for additional reasons. For parents who did not participate in NBS in 2020, the additional reason 'because of COVID' was added.

**Informed decision to participate or not in NBS.** Informed decision was assessed based on the Multidimensional Measure of Informed Choice (MMIC) [23], and a previous study [20, 21], combining knowledge, attitude, and participation in NBS. Knowledge was measured using six items (see S1 Table) with three answer options (true, not true, don't know). Five items with a five-point semantic differential scale were used to measure the respondents' attitude towards NBS (bad–good; useless–useful; not reassuring–reassuring; scary–not scary; annoying–pleasant) (see S2 Table). The attitude items were scored from 1 to 5, where a score of 1 means a negative attitude and a score of 5 a positive attitude.

Knowledge was defined as sufficient when four or more items were answered correctly ($\geq 4$ of 6 items). The five attitude items (scale 1 to 5) were combined into a mean and recoded into an attitude score ranging from 2 (all 5 items very negative; $(1/5)^*10 = 2$) to 10 (all 5 items very positive; $(5/5)^*10 = 10$) score. The attitude of respondents was defined as positive when their attitude score was 6 or more ($\geq 6$ out of 10) and a negative attitude was defined by an attitude score less than six ($< 6$ out of 10).

A well-informed decision was defined based on the MMIC [23]. Parents who made a well-informed decision to participate in NBS were defined as having sufficient knowledge ($\geq 4$ items correctly answered) and a positive attitude (attitude score $\geq 6$) towards NBS. Parents who made a well-informed decision not to participate in NBS were defined as having sufficient knowledge ($\geq 4$ items correctly answered) and a negative attitude (attitude score $< 6$) towards NBS.

In addition to informed choice, three statements were added to assess whether parents made an undisputed choice whether or not to participate in NBS (see S3 Table).

**Views on current NBS.** Views on current NBS consisted of five statements of parents' experiences with the heel prick screening process and the information that was provided on a 5-point Likert scale. Moreover, three questions assessed respondents' perceived reliability of the test, regret, and whether they would participate again on a 5-point answering scale.

**Views on expanding NBS.** Views on expanding NBS consisted of descriptions of seven conditions (e.g. a condition which will give your child symptoms only later in life) about which parents who had participated in NBS were asked if it would be a very bad, bad, neutral, good or very good idea to add these conditions to NBS. In addition, a scenario for including ALD in the NBS program ("ALD scenario"), describing the difference in disease manifestations between males and females, was added (see S1 Appendix; question B.3). Parents were asked whether they thought it was a good idea to test for this condition in NBS, and if so, whether either only boys or both boys and girls should be tested, and why (open text field).

**Background characteristics.** The background characteristics consisted of questions about whether mother, father or both completed the questionnaire, age, marital status, and educational level of the respondent, parents' country of birth, whether or not parents had other children, the sex of the newborn, the result of the NBS (only for NBS participants), whether parents regard alternative medicine or lifestyle (e.g. anthroposophy, homeopathy or natural medicine) as important, whether parents were actively religious and whether parents were planning to have their newborn vaccinated.

## Statistical analyses

The frequencies or means and standard deviations were calculated using descriptive analyses.

Five-point scales were reduced to three categories (e.g. (completely) disagree/neutral/ (completely) agree) to avoid empty or small cells. Differences between the respondents who participated in NBS and those who declined NBS were tested with Chi square tests, or Chi square tests for trend for ordinal categories such as single items on a 5-point scale. Non-parametric Mann-Whitney U tests were done to compare respondents who participated in NBS and those who declined NBS, and also to compare low-/middle- and high- educated parents that participated in NBS, on their attitudes (recoded into a score ranging from 2 (all negative attitudes) to 10 (all positive attitudes)) and knowledge (also recoded into a score ranging from 0 (none correct) to 10 (all correct))/. The comparison based on education, was only done within the group of NBS participants because the numbers of low-educated parents were small (n = 15) in the smaller group of parents that did not participate in NBS (n = 48). Answers to open questions were coded and categorized by one person (MvE) and were discussed with a second person (SW) until agreement was reached. All analyses were performed using SPSS version 25 for Windows (IBM Statistics for Windows, IBM, NY, USA).

## Results

### Respondents' characteristics

In total, 804 of 6,051 (13%) invited parents who participated in NBS completed the questionnaire. Among parents who declined NBS in 2019 and 2020, the questionnaire was completed by 3% (16/515) and 5% (32/647) respectively, meaning that 48 parents who declined participation in NBS were included in the analyses. The characteristics of parents who completed the questionnaire are shown in Table 1. Respondents who declined NBS were more often fathers (19% vs. 8%, p = .03), more often had more than one child (66% vs. 45%, p = .005), were more often actively religious than participants (57% vs. 23%, p < .001), considered alternative medicine/lifestyle as more important more often (67% vs. 21%, p < .001), and were less often inclined to vaccinate their child for infectious diseases (15% vs 96%, p < .001), compared to respondents who participated in NBS. Respondents were higher educated and more often born in the Netherlands, compared to the general Dutch population (2019).

### Reasons to participate in NBS or decline NBS

Table 2 shows that the most important reasons to participate in NBS screening was that disease symptoms in the child can be prevented and because respondents were confident that the results of the heel prick test are reliable. The most important reasons for respondents not to participate were due to their view of life, because they think that the heel prick test is painful for the child, and how their data and their child's data are handled. Among the 32 parents who declined NBS in 2020 (COVID-19 pandemic), two (6%) respondents reported that the COVID-19 virus was an important or very important reason to decline NBS. The open question where respondents described their reasoning in their own words did not reveal any additional reasons.

### Informed decision to participate or not in NBS

Table 3 shows that 83% of respondents that participated in NBS made an informed decision to participate, based on 86% of respondents that showed sufficient knowledge and 97% that had a positive attitude towards NBS. Of the respondents that declined participation in NBS, 44% made an informed choice with sufficient knowledge (62%) and a negative attitude (71%). The difference in the percentage that made an informed decision to participate in NBS between respondents who participated in NBS (83%) and those who declined NBS (44%) was

**Table 1. Characteristics of respondents who participated in NBS participants or declined NBS, compared to the general Dutch population.**

| | Respondents who participated in NBS n = 804[a] | Respondents who declined NBS n = 48[b] | Dutch population 2019[c] |
|---|---|---|---|
| Questionnaire completed by n (%) | | | |
| Mother | 657 (82) | 34 (71) * | |
| Father/partner | 62 (8) | 9 (19) | |
| Both | 85 (11) | 5 (10) | |
| Age of respondent, mean (SD) | 32.2 (4.5) | 32.1 (4.8) | 31.5 |
| Marital status, n (%) | | | |
| Married/registered partnership | 504 (63) | 35 (73) | (58) |
| Living together | 275 (34) | 10 (21) | |
| Single | 23 (3) | 3 (6) | |
| Educational level of respondent[d], n (%) | | | |
| Low/middle | 202 (25) | 15 (33) | (46) |
| High | 602 (75) | 31 (67) | (54) |
| Country of birth mother[e], n (%) | | | |
| Netherlands | 739 (92) | 41 (87) | (67) |
| Western | 40 (5) | 2 (4) | (11) |
| Non-Western | 25 (3) | 4 (9) | (22) |
| Having other children (yes), n (%) | 360 (45) | 31 (66) * | |
| Sex newborn[f], n (%) | | | |
| Male | 413 (53) | 23 (48) | (51) |
| Female | 373 (48) | 25 (52) | (49) |
| NBS (self-reported) result, n (%) | | | |
| Normal | 776 (97) | n.a. | |
| Inconclusive/abnormal[g] | 27 (3) | n.a. | |
| Unknown | 1 (0.1) | | |
| Religion, n (%) | | | |
| Not (actively) religious | 617 (77) | 19 (43) * | |
| (Scarcely) actively religious | 187 (23) | 25 (57) | |
| Alternative medicine/lifestyle is important, n (%) | | | |
| Yes[h] | 166 (21) | 31 (67) * | |
| No | 629 (79) | 15 (33) | |
| Vaccinations, n (%) | | | |
| All vaccinations | 775 (96) | 7 (15) * | |
| Not all vaccinations | 17 (2) | 8 (17) | |
| No vaccinations | 7 (1) | 27 (57) | |
| Vaccinations postponed till later/undecided | 5 (1) | 5 (11) | |

n.a. = not applicable.

The percentage does not always add up to 100% due to rounding.

*. p-value < .05 of difference between respondents participated in NBS and respondents who declined NBS.

[a]The n slightly differed for some of the characteristics due to missing values (n = 795 to 802).

[b]The n slightly differed for some of the characteristics due to missing values (n = 44 to 47).

[c]Dutch population in 2019; for age, marital status, migration background and sex of newborn within all newborns or mothers of newborns in 2019; for educational level within women aged 25–35 years in 2019 (Statline Statistics Netherlands) [24]. Not all data were available, therefore some cells remain empty.

[d]Low education level = Elementary school, lower level of secondary school, and lower vocational training. Middle education level = higher level of secondary school and intermediate vocational training. High education level = High vocational training and university.

[e]Country of birth of mother was categorized as Dutch, Other Western or Non-Western by the following algorithm: Dutch if the mother was born in the Netherlands; Other Western if the mother was born in Europe (excluding Turkey), North America, Oceania, Indonesia or Japan; and Non-Western if the mother was born in Africa, Latin America, Asia (excluding Indonesia and Japan) or Turkey.

[f]Sex could not be specified for 17 twins and 1 triplet.

[g]Inconclusive result (n = 13), abnormal result (n = 12), carrier (n = 1), waiting for an extra heel prick test (n = 1).

[h]Respondents indicated that one or more of the following was important to them: anthroposophy, homeopathy, natural medicine.

**Table 2. Reasons to participate in NBS or decline NBS.**

| "You did or did not participate in the heel prick test. Indicate how important the following reasons were for you in making a decision about participation" | Respondents who participated in NBS | "You did or did not participate in the heel prick test. Indicate how important the following reasons were for you in making a decision about participation" | Respondents who declined NBS |
|---|---|---|---|
| Most important reasons first: | n = 804 Mean (SD) | Most important reasons first: | n = 48 Mean (SD) |
| Because I can prevent my child from getting health complaints from a disorder | 4.62 (0.65) | Because of my view of life (e.g. anthroposophical or naturopathic) | 3.66 (1.24) |
| Because I am confident that the results of the heel prick test are reliable | 4.16 (0.74) | Because I think the heel prick test is painful for my child | 3.40 (1.44) |
| Because the heel prick test reassures me | 4.06 (0.82) | How my data and my child's data are handled | 3.38 (1.41) |
| How my data and my child's data are handled | 2.85 (1.17) | Because I can prevent my child from getting health complaints from a disorder | 2.74 (1.26) |
| Because the government arranges and pays for the heel prick test | 2.83 (1.20) | Because I am confident that the results of the heel prick test are reliable | 2.64 (1.13) |
| Because I think the heel prick test is painful for my child | 2.13 (0.91) | Because the government arranges and pays for the heel prick test | 2.57 (1.12) |
| Because of my view of life (e.g. anthroposophical or naturopathic) | 1.65 (0.98) | Because of my faith or religion | 2.54 (1.61) |
| Because of my faith or religion | 1.36 (0.74) | Because the heel prick test reassures me | 1.93 (1.02) |
| Because of Corona virus[b] | n.a. | Because of Corona virus[b] | 1.74 (1.06) |

n.a. = not applicable.

[a] 1 = not at all important—5 = very important. Maximum of 2 missing values per reason.

[b] Only assessed among non-participants in 2020.

statistically significant (p < .001). S1 and S2 Tables show respondents' answers per knowledge item and per attitude item, respectively, and show significant differences in knowledge-score (7.72 versus 5.87, p < .001) and attitude-score (8.33 vs 5.16, p < .001) between respondents who participated in NBS and those who declined NBS. S3 Table shows that respondents who participated in NBS made a more undisputed choice to participate (p < .001), while NBS non-participants indicated having more doubt (p < .001). S4 Table shows that of all respondents who participated in NBS, respondents with a high educational level had more knowledge

**Table 3. Informed decision to participate in NBS or decline NBS.**

| Knowledge & Attitude | Respondents who participated in NBS n = 801[a] | Respondents who declined NBS n = 45[a] |
|---|---|---|
| | n (%) | n (%) |
| Sufficient knowledge & positive attitude | 667 (83)[b] | 8 (18) |
| Sufficient knowledge & negative attitude | 20 (3) | 20 (44)[b] |
| Insufficient knowledge & positive attitude | 106 (13) | 5 (11) |
| Insufficient knowledge & negative attitude | 8 (1) | 12 (27) |

[a]Three participants and three non-participants completed 3 or less of the 5 attitude questions and were excluded from analyses.

[b]Informed decision (for NBS participants: sufficient knowledge & positive attitude, for NBS non-participants: sufficient knowledge & negative attitude).

about NBS (difference of 0.75, p<0.001) and a more positive attitude towards NBS (difference of 0.46, p<0.001), compared to respondents with a low/middle educational level.

## Parents' views on current NBS

Table 4 shows that 92% of respondents who participated in NBS did not feel pressured by the person who carried out the heel prick. The majority (74%) reported to have received sufficient information about the heel prick test to make a decision about whether or not to participate. Overall, 47% reported that they were well informed about NBS during pregnancy. Most respondents (92%) trust that the government will inform them adequately about the screening results. Most respondents think that the heel prick is reliable (97%), and four respondents (0.5%) reported regretting NBS participation; one parent's child was diagnosed with a condition after NBS, one parent was already in doubt about participating in NBS and did not get satisfactory answers to their questions about NBS, and the other two parents left no remarks on the questionnaire. Almost all respondents (99.6%) would participate again in the case of a future pregnancy.

## Parents' views on expanding Dutch NBS

Most NBS participants (95%) thought it was a good idea to add new conditions to the NBS screening, whereas 4% did not know whether it was a good idea. Only two respondents thought it was a bad idea. Table 5 shows the degree of acceptance of specific conditions about which parents were asked if they thought it was a good idea to add these conditions to the heel prick test. For each condition, the majority of respondents thought it was a good or a very good idea, varying from 59% to 86% agreement. On average, the respondents scored the highest on: 'a condition in which the result could indicate that the child is not sick but that the

**Table 4. Parents' views on current Dutch newborn bloodspot screening.**

| | Participants n = 804 | (Completely) disagree | Neutral | (Completely) agree |
|---|---|---|---|---|
| | Mean (SD) | n (%) | n (%) | n (%) |
| Do you agree with the following statements?[a] | | | | |
| • The person who carried out the heel prick test did not put me under any pressure | 4.35 (0.80) | 27 (3) | 39 (5) | 738 (92) |
| • The person who carried out the heel prick test, was very professional | 4.30 (0.73) | 21 (3) | 47 (6) | 736 (92) |
| • The person who carried out the heel prick test, gave adequate information | 4.05 (0.85) | 50 (6) | 85 (11) | 669 (83) |
| • I was well informed about the heel prick test during pregnancy | 3.11 (1.22) | 273 (34) | 157 (20) | 374 (47) |
| • I received sufficient information about the heel prick test so that I could make a well-founded decision about whether or not to participate | 3.89 (1.05) | 87 (11) | 124 (15) | 593 (74) |
| • I trust that the government will inform me adequately about the heel prick test | 4.34 (0.74) | 21 (3) | 40 (5) | 743 (92) |
| | | Not reliable | Neutral | Reliable |
| How reliable do you think the heel prick is?[b] | 4.49 (0.57) | 2 (0.2) | 22 (3) | 780 (97) |
| | | A lot of regret | Neutral | No regret |
| Do you regret participating in the heel prick?[c] | 4.95 (0.31) | 4 (0.5) | 0 (0) | 800 (99.5) |
| | | Definitely not | Neutral | Definitely |
| If you are ever pregnant again, would you participate in the heel prick?[d] | 4.95 (0.27) | 2 (0.3) | 1 (0.1) | 801 (99.6) |

[a](1 = completely disagree– 5 = completely agree).

[b](1 = not reliable– 5 = reliable).

[c](1 = a lot of regret– 5 = no regret).

[d](1 = definitely not– 5 = definitely).

**Table 5. Degree of acceptance of adding a specific condition to the heel prick test.**

| A condition: | Mean (SD) | (Very) bad idea n (%) | Neutral n (%) | (Very) good idea n (%) |
|---|---|---|---|---|
| for which the result could indicate that the child is not sick but that the mother is sick. | 4.05 (0.72) | 34 (4) | 75 (9) | 695 (86) |
| which will give your child symptoms only later in life. | 4.01 (0.70) | 30 (4) | 102 (13) | 672 (84) |
| for which some of your child's DNA (the genetic material) needs to be tested. | 3.94 (0.76) | 39 (5) | 125 (15) | 640 (80) |
| for which treatment can cause serious side effects for your child. | 3.80 (0.81) | 48 (6) | 204 (25) | 552 (69) |
| with which your child can lead a normal life, such as going to school, participate in sports etc., even without treatment. | 3.75 (0.97) | 107 (13) | 128 (16) | 569 (71) |
| for which it is uncertain whether your child will get any complaints. | 3.58 (0.90) | 104 (13) | 223 (28) | 477 (59) |
| for which there is no treatment or no medication. | 3.56 (0.98) | 121 (15) | 193 (24) | 490 (61) |

mother is sick' (mean: 4.05; SD: 0.72) and the lowest on: 'a condition for which there is no treatment or no medication' (mean: 3.56; SD: 0.98). A minority (5%) believed it was a bad or very bad idea to include disorders for which the child's DNA needs to be tested.

With regard to the "adrenoleukodystrophy (ALD) scenario" (see S1 Appendix, question B.3), only 8 out of 804 respondents (1%) did not think it was a good idea to add a condition to NBS with great differences in disease manifestation and treatability between males and females. Of the 796 respondents that thought it was a good idea to add ALD, a majority of respondents (665/796, 84%) thought it is a good idea to test for this disorder in both boys and girls. Respondents' main reason was "the opportunity to know what to expect" (192/665, 29%), and "because the information is important for both sexes" (105/665, 16%). Only 131/796 respondents (16%) would test only boys because "boys have treatment options and girls do not have treatment options" (80/131, 61%), "consequences for boys are greater" (27/131, 21%), "girls can experience unnecessary stress or worries" (24/131, 18%).

## Discussion

Overall, NBS participants in this study carried out in the Netherlands were very positive about the current NBS program, while those who declined NBS felt differently, and the large majority of NBS participants was also positive about expanding the number of disorders screened for. The proportion of respondents making an informed decision to participate in NBS was lower for NBS non-participants compared to NBS participants, mostly due to a lack of knowledge among NBS non-participants.

### Reasons to participate or not in NBS

The reason respondents reported as most important for participating in NBS was to prevent disease symptoms in their child, in line with the aim of screening [2], as well as with findings from other studies assessing parents' views [25, 26]. For example, one interview study in Australia found that the majority of women with newborns thought screening was useful if it prevents a disease (85%) or reduces severity of a disease (87%) [25]. In an interview study with parents in the UK, the ability to act on screening results was also noted as important, with earlier knowledge on potential conditions in a child allowing for treatment strategies [26].

For parents that declined NBS, the highest rated reasons for declining NBS concerned their view of life, and because they thought that the heel prick test would be painful for their child. Parents who declined NBS were more often in doubt concerning whether to participate or not, compared to NBS participants. In the literature, studies including parents that declined NBS are scarce [27]. A Canadian qualitative study showed that decisions among three declining

parents were shaped by their previous experiences with blood collection [11]. In that same interview study, healthcare professionals reflected on why parents reported to them that they did not wish to participate in NBS. Reasons included: to avoid pain for the baby, previous experiences with healthcare, perceiving the test as unnecessary, religious reasons, and not trusting the government with their child's DNA [11]. Our study suggests that how the parents' data and the child's data are handled might play a role in parents' decision to decline NBS.

Our results showed that parents who declined NBS were more actively religious and more often indicated that alternative medicine/lifestyle is important to them when compared to participating parents. Moreover, more than half of the parents that declined NBS indicated that they were not planning to vaccinate their child for childhood infectious diseases. Declining both the NBS and the vaccination program might therefore also be representative decisions for a way of life. One parent in our study stated that their main reason for declining NBS was 'we live in trust and not in fear' in the open-ended section of the question. Among the parents who declined NBS in 2020, when the COVID-19 pandemic might have caused additional concerns for parents, very few parents (n = 2; 6%) reported that the COVID-19 virus was an important reason to decline NBS.

## Informed participation for current NBS

In the Netherlands, NBS is voluntary, however participation rates are nevertheless very high: above 99% [9]. In the Netherlands, parents are asked for verbal consent before the heel prick is carried out. According to the informed consent procedure [8], the screener should check beforehand whether the parent has both received the information leaflet on NBS screening and read it. If this is not the case, the screener should explain the main points of the screening to the parents. In some other countries, although informed (verbal) consent is likewise required for participation (e.g. United Kingdom) [28], a sense of routinization has been described in which parents did not feel they made an active choice or did not remember giving consent [11]. In the Netherlands, information leaflets on NBS screening are handed out on several occasions, including during pregnancy. However, leaflets are easily overlooked by parents [20, 21]. Given such circumstances of information provision, it has been questioned whether participants perceive participating in NBS as an actual choice [29]. Nevertheless, we have shown that most respondents believe that they were informed well enough to be able to decide to participate, although only 47% of participants said they were well-informed during pregnancy. Previous calls for information provision during pregnancy have been made, as this may allow parents to better absorb and understand the NBS process and the possible implications [13].

Our results indicate that the majority (83%) of the NBS participants made an informed decision, which is comparable to results of a similar study in 2008 (84%) carried out after the expansion of NBS screening from three to seventeen disorders [20, 21]. These results suggest that even with a greater number of conditions included in NBS, the proportion of parents making an informed decision to accept NBS remained the same. The current study found a lower percentage (44%) of informed choice in parents that declined NBS, mostly due to insufficient knowledge about NBS. For example, only 56% of declining parents were aware that conditions tested for with NBS can have serious consequences if untreated (S1 Table). This could indicate room for improvement in the information provision, by explaining or highlighting these consequences. A literature review showed that the main acceptability component identified for newborn screening and other screening interventions related to parental knowledge and understanding [27]. It remains to be seen whether improved information, or other factors such as timing of information provision, will increase knowledge and thus the willingness to

participate among the parents who declined. On the one hand, our results indicate that parents mostly decline because of their beliefs, namely their way of life, religion or beliefs about pain. On the other hand, parents who did decline NBS also indicated having more doubt and a less undisputed choice not to participate.

## Parents' views on expanding Dutch NBS

Our results suggest that NBS participants are generally positive towards the expansion of NBS. Even for conditions for which there is no treatment or medication available or a condition in which it is uncertain whether the child will develop complaints, more than half of the participants thought inclusion of these disorders in the NBS program would be a good idea. This is also supported by earlier research [6, 18, 19]. For example, a Canadian survey study showed that only 38% of parents believed that NBS should only be available for health conditions for which treatment or even supportive treatment is available [6]. Ataxia Telangiectasia (AT) is a specific example of an untreatable condition which occurs as an incidental finding in NBS when screening for severe combined immunodeficiency (SCID). A Dutch survey study found that the most important arguments given by the majority of parents that supported early detection of AT were to prevent a long period of uncertainty between first symptoms and diagnosis (often referred to as a "diagnostic odyssey") and immediate optimal guidance when first symptoms occur [30].

In the United States, the public is generally positive about NBS expansion [5], despite a persistent lack of knowledge about NBS [31]. Our interview study among NBS stakeholders including healthcare professionals, NBS policy makers, researchers, laboratory specialists, and parents in the Netherlands showed that they had a positive attitude toward NBS expansion provided that it is aimed at detecting treatable disorders and achieving health gain [14]. Stakeholders with a "targeted-scope perspective" thought that the focus should be on achieving health gain for the newborn, while other stakeholders expressed broader aims and beneficiaries of NBS, for example also taking reproductive choices for family members into account [14]. Among health professionals, the topic of benefits to family members has been controversial. For example, United States research on healthcare practitioners views on Pompe disease showed that opinions on the use of reproductive benefit as a justification for NBS were markedly divided [32].

In 2021 a pilot study was carried out in four provinces in the Netherlands in which only boys were screened for ALD [33]. For this disorder, the Dutch Health Council advised screening only boys, because only boys are at risk of developing adrenal insufficiency or cerebral ALD for which treatment is available, whereas symptoms in girls usually only develop in adulthood and are untreatable [34]. In our earlier Dutch interview study, we showed that most parents did not seem to perceive the screening of subgroups as problematic in itself [14]. However, the current study showed that most parents that participated in NBS believed that both boys and girls should be tested for a disorder such as ALD, as is the case in the United States [35]. Our findings also showed that these parents support screening for a condition that may not cause symptoms until later in a child's life.

Our study showed that 86% of NBS participants would support the addition of a disorder where NBS reveals that the mother has a certain condition as opposed to the newborn (such as OCTN2 deficiency). Most NBS participants did not object to including conditions in NBS for which DNA needs to be tested. Although many obstacles still need to be addressed before Next Generation Sequencing (NGS) can be implemented in practice, it is likely that NGS will be increasingly applied in NBS [36]. A recent UK public dialogue showed support for the potential use of whole genome sequencing in NBS provided that appropriate resources and

safeguards are in place [37]. Additionally, 60 of 66 (91%) US parents indicated they definitely wanted genomic sequencing in general for their newborn if presented with this option [38].

## Strengths and limitations

A significant strength of this study pertains to the inclusion of parents who declined NBS. Due to the high uptake of NBS in general, non-participants are low in numbers. Parents that declined NBS also tend to decline participating in studies on NBS, while including non-participants' views and attitudes can be valuable [26].

This study also has some limitations. Due to privacy and data protection issues, only the RIVM could invite the participants to complete the questionnaire. This also meant that no reminders could be sent. Due to the low numbers of NBS non-participants, analytical tests between NBS participants and non-participants were not possible. It is possible that respondents were more positive or more negative based on their experiences than parents in general (potential selection bias). Furthermore, the majority of NBS-participating parents were highly educated and of Dutch origin, which may have led to an overestimation of the proportion of parents making an informed decision. Additional subgroup analyses within NBS participants showed that higher educated parents have more knowledge and are more positive about NBS (S4 Table). More research is needed among parents with lower education levels and of non-Dutch origin. Also, different NBS test results (i.e. negative, inconclusive, abnormal) could influence views and attitudes toward the NBS screening. However, in this study only few parents with abnormal and inconclusive test results responded; subgroup analyses for these groups were therefore not possible. More insight into their perceptions is needed—also regarding NBS expansion—as screening for more disorders will result in more abnormal results, including false positives. Finally, parents might not oversee all ethical, social, and financial implications of NBS expansion. Qualitative research methodologies such as in-depth interviews and focus groups give more room for explanation and are more suitable to further explore motivations behind parents' responses in detail [14, 39].

## Conclusions

In conclusion, most NBS participants made an informed decision to participate in NBS and are positive about screening for more conditions. With new conditions on the horizon and fast-developing screening technologies, insights into parents' views on expansion of NBS can help to ensure a program that suits the population needs while safeguarding the ethical principles for screening. In addition to participants' experiences and choices, this study also attempted to provide insight into non-participant attitudes and reasons for declining. These insights may be especially relevant when considering expansion of NBS in order to maintain a high uptake rate (and thus benefit from the health gains of NBS) and provide parents with the information and support they need to make an informed decision that is in line with their values.

## Supporting information

**S1 Appendix. Questionnaire used in this study.**
(DOCX)

**S1 Table. Number of respondents (participants and non-participants in NBS) who correctly answered the knowledge questions.**
(DOCX)

**S2 Table. Attitude towards NBS screening by participation.**
(DOCX)

**S3 Table. Undisputed choice to participate or not in NBS.**
(DOCX)

**S4 Table. NBS knowledge and attitude of low-/middle- versus high-educated parents that participated in NBS.**
(DOCX)

## Acknowledgments

We would like to thank all parents for completing the questionnaire. The National Institute for Public Health and the Environment (RIVM) is acknowledged for sending the invitational letters to the parents, especially Ellen Elsinghorst and Roger Venema. Merel van Eick is thanked for coding the open-ended questions. We thank all project advisors of the PANDA study for their input.

## Author Contributions

**Conceptualization:** Sylvia M. van der Pal, Adriana Kater-Kuipers, Catharina P. B. van der Ploeg, Suze M. P. J. Jans, Stephan Kemp, Rendelien K. Verschoof-Puite, Lion J. M. van den Bosch, Lidewij Henneman.

**Data curation:** Sylvia M. van der Pal, Sophie Wins.

**Formal analysis:** Sylvia M. van der Pal, Sophie Wins.

**Funding acquisition:** Sylvia M. van der Pal, Catharina P. B. van der Ploeg, Rendelien K. Verschoof-Puite, Lion J. M. van den Bosch, Lidewij Henneman.

**Investigation:** Sylvia M. van der Pal, Sophie Wins, Adriana Kater-Kuipers, Suze M. P. J. Jans.

**Methodology:** Sylvia M. van der Pal, Sophie Wins, Tessa van Dijk, Adriana Kater-Kuipers, Catharina P. B. van der Ploeg, Suze M. P. J. Jans, Stephan Kemp, Rendelien K. Verschoof-Puite, Lion J. M. van den Bosch, Lidewij Henneman.

**Project administration:** Sylvia M. van der Pal, Sophie Wins, Tessa van Dijk, Adriana Kater-Kuipers.

**Supervision:** Sylvia M. van der Pal, Lidewij Henneman.

**Writing – original draft:** Sylvia M. van der Pal, Sophie Wins, Jasmijn E. Klapwijk, Tessa van Dijk, Adriana Kater-Kuipers, Lidewij Henneman.

**Writing – review & editing:** Sylvia M. van der Pal, Sophie Wins, Jasmijn E. Klapwijk, Tessa van Dijk, Adriana Kater-Kuipers, Catharina P. B. van der Ploeg, Suze M. P. J. Jans, Stephan Kemp, Rendelien K. Verschoof-Puite, Lion J. M. van den Bosch, Lidewij Henneman.

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
