## [Decision Letter · Decision Letter 0]

18 Apr 2022

PONE-D-22-02051Parents’ views about newborn bloodspot screening  (non-)participation and expansionPLOS ONE

Dear Dr. van der Pal,

Thank you for submitting your manuscript to PLOS ONE. After careful consideration, we feel that it has merit but does not fully meet PLOS ONE’s publication criteria as it currently stands. In the data availability; "Therefore, we invite you to submit a revised version of the manuscript that addresses the points raised during the review process.Because of privacy reasons, the anonymized data per individual for the items described in the manuscript are available on request.". Authors should specify detalis for data access (e.g., data are available from the XXX Institutional Data Access / Ethics Committee (contact via XXX) for researchers who meet the criteria for access to confidential data). Please, note that a statemnet like "data are available through contacting the authors" is not acceptable. Please, review data availability guidelines of PLOS ONE. Please submit your revised manuscript by Jun 02 2022 11:59PM. If you will need more time than this to complete your revisions, please reply to this message or contact the journal office at plosone@plos.org. Please include the following items when submitting your revised manuscript:A rebuttal letter that responds to each point raised by the academic editor and reviewer(s). You should upload this letter as a separate file labeled 'Response to Reviewers'.A marked-up copy of your manuscript that highlights changes made to the original version. You should upload this as a separate file labeled 'Revised Manuscript with Track Changes'.An unmarked version of your revised paper without tracked changes. You should upload this as a separate file labeled 'Manuscript'.

We look forward to receiving your revised manuscript.

Kind regards,

Elsayed Abdelkreem, MD, PhD

Academic Editor

PLOS ONE

Journal Requirements:

Reviewers' comments:

Reviewer's Responses to Questions

**Comments to the Author**

1. Is the manuscript technically sound, and do the data support the conclusions?

Reviewer #1: Yes

Reviewer #2: Partly

2. Has the statistical analysis been performed appropriately and rigorously? 

Reviewer #1: Yes

Reviewer #2: No

3. Have the authors made all data underlying the findings in their manuscript fully available?

Reviewer #1: Yes

Reviewer #2: Yes

4. Is the manuscript presented in an intelligible fashion and written in standard English?

Reviewer #1: No

Reviewer #2: Yes

5. Review Comments to the Author

Reviewer #1: Thank you for the opportunity to review this paper about parents’ attitudes to newborn screening (NBS) in the Netherlands. Manuscript presents a survey distributed to parents, which accepted or declined participation in NBS to analyse if they made an informed decision about participation and to collect their views on current NBS and its possible expansion. The results are based on a sample of 804 NBS participants and 48 NBS non-participants and might give an important feedback to Dutch NBS programme. However, some changes are needed to clearly present the research conducted and its results.

Title

I would suggest to the authors to either substitute the term (non-)participation with participation or rephrase the title completely.

Introduction

I miss the information about Dutch NBS programme. It would be useful for reader to learn how, when and by whom are parents informed about NBS, if the information is standardised, if there is an informed consent in place, when the sampling is performed, how many disorders are included in the current Dutch NBS, how are parents informed about result, etc. All these elements are also important for parents’ views about NBS. Some of this information is in the Discussion paragraph on Informed participation but reader should learn these facts earlier.

The aims of the study are not well described. I would expect to find their overview in the last paragraph of the introduction section. I have learned only in methods section that survey focused also on informed decision about participation in NBS.

Materials and Methods

Results

Table 1 – Can authors specify what they mean by low/middle and high education?

Table 1 - I guess the information about general Dutch population is not available and that’s why some of the boxes stay empty. Short explanation should be provided even it would be very interesting to know these numbers for comparison.

The presentation of the results in Tables 1, 2, 3, 4, 5 and S1 should be unified. The numbers in columns are sometimes presented with percentages and sometimes the percentage sign is only in the table header and in Table 1 it is also in a different column.

Can you specify better how the questions for non-participants were phrased? I revealed from the heading of the first column in Table 2 that the item in survey sounded like “I decided not to participate because I can prevent my child from getting health complaints from a disorder”. I have the same problem with next two items in Table 2.

On Page 14 in the paragraph on Parents’ views on current NBS survey revealed that few respondents had regretted NBS participation. Did they state why?

Discussion

On page 19 in the first paragraph authors say there finding is in line with other studies but they are not cited.

On the lines 310-311: “Our study suggests that how the data and … are handled…” Can you specify which data?

On page 22 in the second paragraph, can you specify the term stakeholders? Does it mean parents, healthcare professionals, healthcare providers or some other groups involved?

Conclusions

Can you explain the term “Dutch ethical principles for screening”? In my opinion ethical principles for screening are universal at least in Western countries. Does it mean ethical principles recommended by Dutch Health Council or other governance body?

General issue:

A stringent proof read would help to correct typos, grammatical errors, complex and unclear sentences, and non-standard use of parentheses. The last sentence in the conclusions is an example of most of these, but they are throughout the paper.

Reviewer #2: This is an excellent article overall and I particularly appreciate the efforts to recruit and gain insight into reasons for non-participation in NBS!

The introduction and discussion would be enhanced by inclusion of more recent articles in the literature; for example, those which have explored perspectives of parents with children who were diagnosed at NBS for conditions with late-onset subtypes and the ethical quandaries in detection of these at birth (see suggestions below).

Also, analytical statistics need to be performed for study results involving comparisons between groups, in order for them to be accepted as not obtained due to chance (significant). I know that your sample size is low for the non-participating group, but such analysis can be done using certain statistical techniques and vastly increases the conclusions you can draw from your data. Otherwise, there are certain statements / conclusions you just can’t scientifically make. As a result, not all of the conclusions made in this manuscript are supported by the data (yet).

Please see attachment for specific feedback throughout the manuscript.

6. PLOS authors have the option to publish the peer review history of their article (what does this mean?). If published, this will include your full peer review and any attached files.

Reviewer #1: No

Reviewer #2: No

---

## [Author Response · Author response to Decision Letter 0]

16 Jun 2022

[see also the uploaded document with our "response to the reviewers]"

Dear Editor,

We would like to thank the reviewers for providing valuable feedback on our manuscript “Parents’ views on declining newborn bloodspot screening, participation and expansion”. We were glad that the reviewers commented favorably on our paper and have edited our manuscript according to their comments. Below you will find our point-by-point reply to their comments. Changes in the revised manuscript are visible with ‘track changes’.

We look forward to hearing your decision.

Dr. Sylvia van der Pal

Editor comments:

Answer: Format checked next to PLOS ONE requirements and minor changes made (see track changes). Supporting information, titles and file names were adjusted to style requirements (e.g. S1 Table) and upload supporting information as separate files.

2. Please provide additional details regarding participant consent. In the ethics statement in the Methods and online submission information, please ensure that you have specified what type you obtained (for instance, written or verbal, and if verbal, how it was documented and witnessed). 

Answer: Under the header “Study population” (p. 6) we had described the consent procedure in the following way: “Before parents completed the (online) questionnaire they were asked to tick a box to confirm that they had read the information letter and that they gave their informed consent to participate in the study.” We have moved this to the Design and ethics statement section (p. 5). To further specify the type of consent we added “(written consent)” to the sentence.

3. In your Data Availability statement, you have not specified where the minimal data set underlying the results described in your manuscript can be found. 

Answer: The Data Availability statement has been amended to “The data underlying the results are available from the secretary of department of Child Health, TNO (contact via childhealthsecretary@tno.nl) for researchers who meet the criteria for access to confidential data”

Reviewers’ Comments:

Reviewer #1: Thank you for the opportunity to review this paper about parents’ attitudes to newborn screening (NBS) in the Netherlands. Manuscript presents a survey distributed to parents, which accepted or declined participation in NBS to analyse if they made an informed decision about participation and to collect their views on current NBS and its possible expansion. The results are based on a sample of 804 NBS participants and 48 NBS non-participants and might give an important feedback to Dutch NBS programme. However, some changes are needed to clearly present the research conducted and its results.

Title

I would suggest to the authors to either substitute the term (non-)participation with participation or rephrase the title completely.

Answer: The title was changed to: “Parents’ views on accepting, declining, and

expanding newborn bloodspot screening.” 

Introduction

I miss the information about Dutch NBS programme. It would be useful for reader to learn how, when and by whom are parents informed about NBS, if the information is standardised, if there is an informed consent in place, when the sampling is performed, how many disorders are included in the current Dutch NBS, how are parents informed about result, etc. All these elements are also important for parents’ views about NBS. Some of this information is in the Discussion paragraph on Informed participation but reader should learn these facts earlier.

Answer: In the introduction, we added information about the Dutch context. This can be found in the following sentences: “In the Netherlands NBS participation is not mandatory and parents are asked if they agree to NBS [8]. NBS is nevertheless well accepted. In 2020, 99.4% of parents of newborns participated in newborn screening which, in that year, included 24 disorders [9].”

Additionally, we added context to the Method section, under the new header “Setting”: “In the Netherlands, NBS is organized and coordinated by the Dutch National Institute for Public Health and the Environment (RIVM). Standardized information on NBS, including a leaflet, is provided during pregnancy by an obstetric healthcare provider, and issued before the screening is done, a few days after birth, at home by a youth health care worker, maternity nurse or midwife. When the baby is admitted to the hospital during the first week after birth, the blood spot is collected by a hospital health care worker. If the NBS result is good, parents will get a letter from RIVM within five weeks, as from March 1st, 2020 (before that time, no result was given). If an abnormality is detected, the general practitioner will inform parents about the results.”

The aims of the study are not well described. I would expect to find their overview in the last paragraph of the introduction section. I have learned only in methods section that survey focused also on informed decision about participation in NBS.

Answer: The aims of the study were changed to: “this study aimed to assess parents’ reasons for, and views on participating or not participating in Dutch NBS, whether they make an informed decision, and their views on expansion of NBS in the Netherlands.”

Materials and Methods

Results

Table 1 – Can authors specify what they mean by low/middle and high education?

Answer: Added as a footnote below Table 1: “dLow education level = Elementary school, lower level of secondary school, and lower vocational training. Middle education level = higher level of secondary school and intermediate vocational training. High education level = High vocational training and university.” 

Table 1 - I guess the information about general Dutch population is not available and that’s why some of the boxes stay empty. Short explanation should be provided even it would be very interesting to know these numbers for comparison.

Answer: We agree that more information would be interesting, however this is not available for this group. An explanation on the empty boxes was added as a footnote below Table 1, footnote c: “Not all data were available, therefore these cells remain empty.”

The presentation of the results in Tables 1, 2, 3, 4, 5 and S1 should be unified. The numbers in columns are sometimes presented with percentages and sometimes the percentage sign is only in the table header and in Table 1 it is also in a different column.

Answer: For all Tables we removed the percentage signs after the numbers in each relevant cell. In Table 1 the percentage sign is in the rows, because age is included which uses means and SD’s.

Can you specify better how the questions for non-participants were phrased? I revealed from the heading of the first column in Table 2 that the item in survey sounded like “I decided not to participate because I can prevent my child from getting health complaints from a disorder”. I have the same problem with next two items in Table 2.

Answer: Changed the heading of the first column in Table 2 to “You did or did not participate in the heel prick test. Indicate how important the following reasons were for you in making a decision about participation in the heel prick test”, which is how the question was formulated in the questionnaire.

On Page 14 in the paragraph on Parents’ views on current NBS survey revealed that few respondents had regretted NBS participation. Did they state why?

Answer: We did not ask for reasons for regret in the question but we did check whether the four parents that indicated regret had left any remarks on the questionnaire. Two parents did not leave any remarks. One parent, who had ‘a bit regret’, indicated having been in doubt about participation but that they participated because they saw no disadvantages. This parent requested further information about the heritability of the NBS diseases but did not get a satisfactory answer. One other parent did not leave a remark but indicated that NBS showed that their child was diagnosed with one of the diseases. We have now specified this in the manuscript. 

Discussion

On page 19 in the first paragraph authors say there finding is in line with other studies but they are not cited.

Answer: This statement refers to the two interview studies that are summarized thereafter. The references are now shown after this sentence.

On the lines 310-311: “Our study suggests that how the data and … are handled…” Can you specify which data?

Answer: This sentenced referred also to parents’ data, therefore we included this in the sentence: “Our study suggests that how the parents’ data and the child’s data are handled might play a role in parents’ decision to decline NBS. This was a general question about (privacy of) data. We are unable to specify further, because this was not specified in the questionnaire.

On page 22 in the second paragraph, can you specify the term stakeholders? Does it mean parents, healthcare professionals, healthcare providers or some other groups involved?

Answer: We changed the sentence to “Our interview study among NBS stakeholders including healthcare professionals, NBS policy makers, researchers, laboratory specialists and parents in the Netherlands showed that…”

Conclusions

Can you explain the term “Dutch ethical principles for screening”? In my opinion ethical principles for screening are universal at least in Western countries. Does it mean ethical principles recommended by Dutch Health Council or other governance body?

Answer: We agree with the reviewer and removed the word “Dutch”.

General issue:

A stringent proof read would help to correct typos, grammatical errors, complex and unclear sentences, and non-standard use of parentheses. The last sentence in the conclusions is an example of most of these, but they are throughout the paper.

Answer: We followed the reviewers advice and asked a professional translator to proofread the manuscript and correct mistakes. These corrections can be seen via ‘track changes’ throughout the manuscript.

Reviewer #2

Overall

This is an excellent article overall and I particularly appreciate the efforts to recruit and gain insight into reasons for non-participation in NBS!

The introduction and discussion would be enhanced by inclusion of more recent articles in the literature; for example, those which have explored perspectives of parents with children who were diagnosed at NBS for conditions with late-onset subtypes and the ethical quandaries in detection of these at birth (see suggestions below).

Also, analytical statistics need to be performed for study results involving comparisons between groups, in order for them to be accepted as not obtained due to chance (significant). I know that your sample size is low for the non-participating group, but such analysis can be done using certain statistical techniques and vastly increases the conclusions you can draw from your data. Otherwise, there are certain statements / conclusions you just can’t scientifically make.

Abstract

Please reword the 1st sentence is the results subsection of the abstract to make clear that these are the most important reasons reported by the parents in the study.

Answer: We amended the sentence to “The most important reason for parents to participate in NBS was to prevent health complaints”.

You cannot make the comparisons between groups in the 2nd and 3rd sentences of the abstract results without having analytical statistics to back up your claim. 

Answer: We have now added the according p-values to the abstract. 

In the final section of the abstract results, please reword to say that “Most parents in the NBS-participating group agreed……”

Answer: We replaced “parents” with “NBS participants”

In the abstract conclusions section, please reword to say that “Most parents in the NBS-participating group made an informed decision……” Might include a sentence about the non-participating group’s informed decisions not to participate as well, since this is new data and relevant to your results.

Answer: Added “NBS” to participants. 

Introduction

The van Dijk article citation is excellent, but a lot of the other literature cited on parental perspectives on NBS in the Introduction is dated. Recent articles have been published exploring the perspective of parents whose children’s conditions were actually detected via NBS (e.g. Pruniski et al 2018’s Newborn screening for Pompe disease: Impact on families)(White et al 2021’s Absorbing it all – a meta-ethnography of parents’ unfolding experiences of NBS)(Kariyawasam et al 2021’s We needed this)(etc.). In addition, a few recent articles have asked adults with genetic conditions whether they would have wanted to know their condition at birth via NBS (e.g. Lisi et al’s 2016 & 2021 articles). Please update your literature review to include some of these.

Answer: Thank you for pointing us to these papers. We have included the review paper of White et al. in the manuscript introduction and discussion, as was suggested by the reviewer. However, besides this review paper, these are all papers about the (psychological) impact on parents and families of a specific abnormal NBS result such as for Pompe disease (Pruniski 2018) or for SMA (Kariyawasam 2021), or on the perspectives of health professionals (Lisi 2016) or of adult patients on NBS inclusion of specific diseases (Lysosomal storage diseases), while our study assesses the general views of a general group of parents on NBS, with very few parents having an abnormal results. In our next study that is currently being done, we focus on psychological well-being after abnormal results and we will include these articles.

Methods

Page 5, lines 95-99 - Are the #’s of parents who declined participating in NBS in 2019 and 2020 from the entire year or just the five-weeks-earlier period the parents who consented to NBS were taken from? It’s not clear from the wording in the text – please clarify.

Answer: We added the words “the whole of” to reflect that these parents were from the entire year. The sentence now reads: “In addition, all parents who declined participation in NBS in the whole of 2019 (n = 515) received an invitation to complete the same online questionnaire at the beginning of 2020, …”

Page 6, line 111 – Please clarify whether the 5 parents the questionnaire was tested on had allowed their children to have NBS or declined it, whether their children had positive NBS results or whether all had received negative NBS results, whether or not the 5 parents also participated in the study, etc.

Answer: We don’t know. The questionnaire was piloted among parents. This was done to see whether the questionnaire was clear. We did not specifically ask or look at their answers and did not incorporate their answers in the data as the questionnaire was in its test phase.

Page 7, lines 148-151 – Tables S1, S2, and S3 allow the reader to see the questions asked of the participant. It would be nice to see the Views on current NBS questions as well. Consider adding a copy of the entire survey in an Appendix for reader reference.

Answer: We added the survey in an Appendix as suggested by the reviewer. In the Method section, under the header Questionnaire and procedure (p.7) we refer to the appendix in the following way: “A copy of the entire questionnaire can be found in S1 Appendix.” The numbering for the other supplemental materials has been adjusted according to order of appearance (e.g. S1 Table is now S2 Table). The ALD scenario formerly described as S4 appendix has been removed, as this is incorporated in S1 Appendix (question B.3).

Page 8, line 153 – Please include which 7 conditions participants were asked to evaluate for expanded NBS, if these questions were about specific genetic conditions. If they are about more general scenarios (as Table 5 suggests), please make that clear. In general, the questions asked seem to be spread out over a large # of tables. Again, consider consolidating them all by placing a copy of the entire survey in an Appendix for ease of reader reference.

Answer: We added an example to illustrate: “Views on expanding NBS consisted of descriptions of seven conditions (e.g. a condition which will give your child symptoms only later in life) for which parents who had participated in NBS were asked if it would be a very bad, bad, neutral, good or very good idea to add these conditions to NBS.” We also added the survey in an Appendix (S1 Appendix) as suggested by the reviewer in the previous comment.

Results

Please clarify the %’s in the “Statistics” column in Table 1. As stated now (%s), I do not know how to interpret them. Are they simply meant to reflect the % of each row out of the total possible? If so, why do we not see them for every row? If they are meant to represent something else, what else? A p-value to determine whether there were any pre-existing significant differences between groups would lend greater credibility to your results.

Answer: We have now replaced the %’s in the last column with (..), as this is indeed the way it is displayed in the other columns. We have now also clarified that this last column represents the values within the Dutch population to put the values in the previous column’s in perspective. As this is mainly for perspective we did not include statistical analyses. We did include statistical analyses to compare the other two columns with respondents who did and did not participate in NBS. It is however questionable whether differences are caused by actual differences between the two groups or a difference in response-bias in the two groups. 

Also in Table 1, please define how you categorized someone as having “low/middle” levels of education versus “high” education.

Answer: Added as a footnote below Table 1: “dLow education level = Elementary school, lower level of secondary school, and lower vocational training. Middle education level = higher level of secondary school and intermediate vocational training. High education level = High vocational training and university.” 

Table 2: Please include p-value or similar analytical statistic to allow reader to determine whether difference between columns is significant or not. 

Answer: We believe that these means cannot be compared as the reasoning for those who participated in NBS and those who declined cannot be compared. To make this more clear we have now redesigned the table with displaying the most important reason for each group at the top. 

Page 13, lines 236-237 – You cannot conclude that respondents who participated in NBS made a more undisputed choice to participate or that NBS non-participants indicated more doubt without some form of analytical analysis to show that the difference in score was not due to chance. Please do analysis, as was done in Table S5 (also, please define the terms “low-middle educated” and “high-educated in Table S5).

Answer: We have performed Chi square tests for ordinal trend on the variables displayed in S4 Table and added this to the table and added a sentence in the text under Informed Decision: “S4 Table shows that respondents who participated in NBS made a more undisputed choice to participate (p<.001), while NBS non-participants indicated having more doubt (p<.001)”. We also added a footnote beneath Table S5: “dLow education level = Elementary school, lower level of secondary school, and lower vocational training. Middle education level = higher level of secondary school and intermediate vocational training. High education level = High vocational training and university.” 

Discussion

Page 18, lines 287-289 - The beginning of the discussion states that overall respondents were very positive about the current NBS program, as well as expanded screening. However, this is confounded by the vastly different sample sizes between those who opted in for NBS vs those that opted out. You can reword this to say that those participants who opted into NBS were very positive about the current NBS system, while those who opted out felt differently (but you have to do analytical statistics to show the differences between groups were not obtained by chance).

Answer: We now did statistical tests and on the knowledge-score and attitude-score differences between NBS participants and those who declined NBS and added this to S2 and S3 tables and added a sentence about this in the Results under Informed Decision; “S2 and S3 Tables show respondents’ answers per knowledge item and per attitude item, respectively, and shows a difference in knowledge-score (7.72 versus 5.87, p<.001) and attitude-score (8.33 vs 5.16, p<.001) between respondents who participated in NBS and those who declined NBS.” Furthermore, we amended this first sentence in the discussion to: “Overall, NBS participants in this study carried out in the Netherlands were very positive about the current NBS program, while those who declined NBS felt differently, and the large majority of NBS participants was also positive about expanding the number of disorders screened for.”

Page 18, lines 289-291 – Likewise, in order to make the statement here regarding the proportion of those making an informed decision being lower in one group than the other, a p-value or other analytical statistic is needed to prove that the results were not obtained by chance. The only part of this statement that has been statistically analyzed is the education part (Table S5).

Answer: We now statistically tested whether the difference in informed decision (yes/no) between respondents that participated in NBS and those who declined NBS with a chi square test and this was highly significant (p<.001). We included this in the Results under Informed Decision: “The difference in the percentage that made an informed decision to participate in NBS between respondents who participated in NBS (83%) and those who declined NBS (44%) was statistically significant (p<.001).” 

Page 19, line 294 – Would recommend rewording this sentence to “The reason respondents reported as most important to participate in NBS was…..,” so as to make clear it is the respondents’ opinion, not the authors. This is well done at the beginning of the 2nd paragraph on this page, so maybe copy that wording for the 1st paragraph as well.

Answer: Amended to “The reason respondents reported as most important for participating in NBS was to prevent disease symptoms in their child, in line with the aim of screening [2], …”

Page 19, lines 303-304 – Cannot say that “Non-participating parents were more actively religious…..compared to participating parents” without a p-value to prove this.

Answer: We now performed statistical tests and included this in Table 1. 

Page 19, lines 312-314 - Cannot say that “Non-participating parents were more often in doubt….compared to NBS participants” without a p-value to prove this.

Answer: We now performed statistical tests and included this in S4 table. 

Page 20, line 339 – It looks as though you have only included the group that made the informed decision to participate in this % (83%). Wouldn’t it be more accurate to include those who made the informed decision NOT to participate in this # as well? Either that or change the wording of the sentence to make clear that you are only referring to the NBS-participating group, not the entire group of participants.

Answer: We changed participating parents to NBS participants to clarify that here we only refer to the group that participated; “Our results indicate that the majority (83%) of the NBS participants made an informed decision, …”. 

We did include those who made an informed decision not to participate. This is shown in the results section and discussion: “The current study found a lower percentage (44%) of informed choice in parents that declined NBS.”

Page 21, line 344 – You need a p-value to be able to scientifically declare this a difference between groups. 

Answer: We have now done a statistical test to compare the %’s of informed decision between respondents who participated in NBS and those who declined NBS and added a sentence about this in the Results under Informed Decision: “The difference in the percentage that made an informed decision to participate in NBS between respondents who participated in NBS (83%) and those who declined NBS (44%) was statistically significant (p<.001).”

Page 21, line 355 - You need a p-value to be able to scientifically declare this a difference between groups. 

Answer: We have now done statistical tests to look at this difference and added this to the S4 table. 

Page 21-23, section on “Parents’ views on expanding Dutch NBS” – This section does a much better job of incorporating some newer article citations – love it A few others which are relevant and add a different perspective include O’Connor et al’s 2018 Psychosocial impact on mothers receiving expanded NBS results, as well as some of the research on NBS for lysosomal storage diseases mentioned earlier in this review. 

Answer: Although we fully agree with the reviewers that some newer citations would be helpful; there are not many recent papers on parents’ views on expanded NBS, moreover, the citations that are suggested by the reviewer, though very interesting, are mainly about the psychological impact of receiving an abnormal result. We therefore believe that these are not completely helpful in discussing our findings that focus more on the views of parents with a normal result. In our next study that is currently being done, we focus on psychological well-being after abnormal results and we will include these articles.

Page 22, lines 376-379 – Not sure who you are including as stakeholders in this section, but you may find it interesting to compare with the results in Davids et al 2021 article asking about the use of NBS for reproductive choices for family members regarding Pompe disease (Health care practitioners’ experienced-based opinions on providing care after a positive NBS for Pompe)

Answer: Thank you for pointing us to the paper of Davids et al. We included this paper in the discussion. “Among health professionals, the topic of benefits to family members has been controversial. For example, United States research on healthcare practitioners views on Pompe disease showed that opinions on the use of reproductive benefit as a justification for NBS were markedly divided.”

Moreover, we have added which stakeholders were included in the interview study as also suggested by Reviewer #1. “Our interview study among NBS stakeholders including healthcare professionals, NBS policy makers, researchers, laboratory specialists, and parents in the Netherlands showed that they had a positive attitude toward NBS”. 

Page 22, lines 390-393 – Please clarify the wording of these sentences to make clear that the percentages you report only pertain to the NBS-participating-parents, not the entire group of respondents.

Answer: Amended by adding “that participated in NBS” to specify which parents, and replaced “parents” by “NBS participants” in the following sentences describing our study results.

Page 24, Conclusions section - Please clarify the wording of the first sentence to make clear that the conclusions you report only pertain to the NBS-participating-parents, not the entire group of respondents. Any other conclusion is confounded by the vast difference is sample size between groups.

Answer: Changed “parents” to “NBS participants”

Supplementary Tables & Figures

Table S1: Please include p-value or similar analytical statistic to allow reader to determine whether difference between columns is significant or not. You might also consider adding a column to show how many participants checked “I don’t know” as opposed to answering incorrectly, since both choices were given to participants.

Answer: We have added the mean Knowledge-score to the S2 table and statistically tested the significance of the difference between respondents who participated in NBS and those who declined NBS. We added a sentence about this in the results under Informed Decision: “S2 and S3 Tables show respondents’ answers per knowledge item and per attitude item, respectively, and show significant differences in knowledge-score (7.72 versus 5.87, p<.001) and attitude-score (8.33 vs 5.16, p<.001) between respondents who participated in NBS and those who declined NBS.” We however feel that emphasizing on the %’s that gave a correct answer provides the reader with a clear overview and therefore decided not to add the participants that answered ‘I don’t know’. 

Table S2: Please include p-value or similar analytical statistic to allow reader to determine whether difference between columns is significant or not. 

Answer: We have added the mean Attitude-score to the S3 table and statistically tested the significance of the difference between respondents who participated in NBS and those who declined NBS. We added a sentence about this in the results under Informed Decision: “S2 and S3 Tables show respondents’ answers per knowledge item and per attitude item, respectively, and show significant differences in knowledge-score (7.72 versus 5.87, p<.001) and attitude-score (8.33 vs 5.16, p<.001) between respondents who participated in NBS and those who declined NBS.” 

Table S3: Please include p-value or similar analytical statistic to allow reader to determine whether difference between columns is significant or not. 

Answer: We now performed statistical tests and added this to the S4 table. We have also added a sentence in the Results under Informed Decision: “S4 Table shows that respondents who participated in NBS made a more undisputed choice to participate (p<.001), while NBS non-participants indicated having more doubt (p<.001).”

---

## [Decision Letter · Decision Letter 1]

22 Jul 2022

Parents’ views on accepting, declining, and expanding newborn bloodspot screening

PONE-D-22-02051R1

Dear Dr. van der Pal,

We’re pleased to inform you that your manuscript has been judged scientifically suitable for publication and will be formally accepted for publication once it meets all outstanding technical requirements.

Kind regards,

Elsayed Abdelkreem, MD, PhD

Academic Editor

PLOS ONE

Additional Editor Comments (optional):

Reviewers' comments:

Reviewer's Responses to Questions

**Comments to the Author**

1. If the authors have adequately addressed your comments raised in a previous round of review and you feel that this manuscript is now acceptable for publication, you may indicate that here to bypass the “Comments to the Author” section, enter your conflict of interest statement in the “Confidential to Editor” section, and submit your "Accept" recommendation.

Reviewer #1: All comments have been addressed

2. Is the manuscript technically sound, and do the data support the conclusions?

Reviewer #1: Yes

3. Has the statistical analysis been performed appropriately and rigorously? 

Reviewer #1: Yes

4. Have the authors made all data underlying the findings in their manuscript fully available?

Reviewer #1: Yes

5. Is the manuscript presented in an intelligible fashion and written in standard English?

Reviewer #1: Yes

6. Review Comments to the Author

Reviewer #1: (No Response)

7. PLOS authors have the option to publish the peer review history of their article (what does this mean?). If published, this will include your full peer review and any attached files.

Reviewer #1: No

---

## [Editor Report · Acceptance letter]

9 Aug 2022

PONE-D-22-02051R1 

Parents’ views on accepting, declining, and expanding newborn bloodspot screening 

Dear Dr. van der Pal:

I'm pleased to inform you that your manuscript has been deemed suitable for publication in PLOS ONE. Congratulations! Your manuscript is now with our production department. 

Kind regards, 

on behalf of

Dr. Elsayed Abdelkreem 

Academic Editor

PLOS ONE